# PatchPilot: A Cost-Efficient Software Engineering Agent with Early Attempts on Formal Verification

Hongwei Li [1]   Yuheng Tang [1]   Shiqi Wang [2]   Wenbo Guo [1]

## Abstract

Recent research builds various patching agents that combine large language models (LLMs) with non-ML tools and achieve promising results on the state-of-the-art (SOTA) software patching benchmark, SWE-bench. Based on how to determine the patching workflows, existing patching agents can be categorized as agent-based planning methods, which rely on LLMs for planning, and rule-based planning methods, which follow a pre-defined workflow. At a high level, agent-based planning methods achieve high patching performance but with a high cost and limited stability. Rule-based planning methods, on the other hand, are more stable and efficient but have key workflow limitations that compromise their patching performance. In this paper, we propose PatchPilot, an agentic patcher that strikes a balance between patching efficacy, stability, and cost-efficiency. PatchPilot proposes a novel rule-based planning workflow with five components: reproduction, localization, generation, validation, and refinement (where refinement is unique to PatchPilot). We introduce novel and customized designs to each component to optimize their effectiveness and efficiency. Through extensive experiments on the SWE-bench benchmarks, PatchPilot shows a superior performance than existing open-source methods while maintaining low cost (less than 1$ per instance) and ensuring higher stability. We also conduct a detailed ablation study to validate the key designs in each component. Our code is available at https://github.com/ucsb-mlsec/PatchPilot.

[1]Department of Computer Science, University of California, Santa Barbara, USA [2]Meta, New York, USA. Correspondence to: Wenbo Guo <henrygwb@ucsb.edu>.

*Proceedings of the 42nd International Conference on Machine Learning*, Vancouver, Canada. PMLR 267, 2025. Copyright 2025 by the author(s).

## 1. Introduction

Automatic patching of issues and vulnerabilities has long been a challenging task in software engineering and security (Jiang et al., 2021; Le Goues et al., 2021; Monperrus, 2018; Gazzola et al., 2018). Before the emergence of generative AI, automated code generation primarily relied on program synthesis (Feng et al., 2018; Huang et al., 2019), which requires human-written specifications and cannot be applied to complex programs due to the constraints of SMT solvers. With the recent success of LLMs in various generative tasks (Peng et al., 2023; Lian et al., 2023; Ghosal et al., 2023; Huang et al., 2024), particularly in code generation (Zhu et al., 2024; Anthropic, 2024; Achiam et al., 2023; Team et al., 2023; Roziere et al., 2023), researchers recently started exploring their applications in automatically fixing software vulnerabilities. They build LLM-based agents that automatically analyze and fix issues in real-world codebases (Wang et al., 2024b; Liu et al., 2024; Ruan et al., 2024; Zhang et al., 2024b; Yang et al., 2024a).

Technically speaking, existing patching agents consist of three main components: localization, generation, and validation. The localization component identifies the code snippets responsible for the issue that need to be fixed. The generation component produces patch candidates, while the validation component selects the final patch from candidates. There are two ways to schedule these components: agent-based planning(Yang et al., 2024a; moatless, 2024; Zhang et al., 2024b; IBM, 2024; Liu et al., 2024; CodeR, 2024; Pedregosa et al., 2011; Ma et al., 2024; Wang et al., 2024b; Amazon, 2024; Zhang et al., 2024a), which utilizes LLMs to dynamically determine the patching workflow for different issues; and rule-based planning(Xia et al., 2024; Ouyang et al., 2024) that follows a fixed, predefined workflow for all issues, as specified by humans. Although achieving high patching performance, agent-based planning methods suffer a high cost and are not stable, which significantly limits their applicability in the real world. In contrast, existing rule-based planning methods are more stable and cost-efficient but have limited patching performance due to limitations in their planning workflows.

In this paper, we present PatchPilot, a novel patching framework that balances the *patching efficacy, stability, and cost-efficiency*. At a high level, PatchPilot designs a rule-based

planning workflow composed of five components: reproduction, localization, generation, validation, and refinement. Given an issue as input, PatchPilot first reproduces the issue and retrieves related testing cases and finds the root cause (code snippets causing the issue) through localization. Its generation and validation components then generate patch candidates and validate whether they fix the issues while preserving the benign functionalities. Unlike existing rule-based patchers(Xia et al., 2024), which regenerate patches from scratch whenever validation fails, PatchPilot introduces a novel refinement component that iteratively improves the current patch based on validation feedback. This refinement continues until the patch passes validation (i.e., becomes "qualified") or the iteration limit is reached. This design is aligned with the human patching workflow, which requires multiple rounds of trials and errors.

As specified in Section 3, each component has its own technical challenges, and we introduce specified designs to address them. Specifically, first, we not only reproduce the issue but also find related benign tests, which later are critical for determining whether the generated patches break normal functionalities during the validation. Second, we design our localization to provide not only the root cause but also the related context that is necessary for patching, and design additional tools for localization to retrieve necessary information from the codebase. Third, for generation, we break it down into patch planning, which produces a multi-step patching plan, and patch generation, which generates patches following the plan. This design is inspired by the recent emergence of inference-phase reasoning (Wei et al., 2022; Yao et al., 2024; Yang et al., 2024b). Having a detailed plan can explicitly prompt the LLM to think deeper and reason about the issue and give more comprehensive patching solutions, which is more effective than directly prompting the model to generate the whole patch.

Through extensive evaluations, we first show that PatchPilot outperforms all SOTA open-source methods on the SWE-bench Lite and SWE-bench Verified benchmark (Jimenez et al., 2023). Besides, we show that PatchPilot achieves the lowest cost among both top open-source and closed-source methods, validating its balance in patching accuracy and cost-efficiency. Second, we demonstrate that PatchPilot is more stable than the SOTA agent-based planning method OpenHands (Wang et al., 2024b), validating the advantage of rule-based planning in terms of stability. Finally, we validate the key designs discussed above through a detailed ablation study and demonstrate that PatchPilot is compatible with multiple SOTA LLMs. Although PatchPilot is not on top of the SWE-bench, to the best of our knowledge, it achieves the best balance between patching performance, stability, and cost-efficiency. These are critical for practicality, making PatchPilot a promising candidate for deployment in real-world scenarios.

**Early attempts on verifiable patching.** As detailed in section 5, we further add a formal verification component where we leverage LLM to generate specifications and use the Z3 solver to provide a formal guarantee on patch correctness. Although we only verify eleven patches in SWE-bench Lite, this marks an early exploration of verifiable patching agents.

## 2. Existing Patching Agent and Limitations

At a high level, existing patching agents mainly have three components: localization, generation, and validation. The *localization* component pinpoints the code snippets that cause the issue and need to be fixed (denoted as "root cause"), the *generation* produces patch candidates, and the *validation* tries to find a final patch in the candidates. Although they have similar components, based on planning strategies, existing patching agents can be categorized into *agent-based planning* and *rule-based planning*. Agent-based planning leverages LLMs to determine the patching workflow (i.e., deciding when and which components to call), which can be different from different issues. On the contrary, rule-based planning follows a fixed workflow for all issues pre-specified by humans.

**Agent-based planning.** Most existing patching agents follow agent-based planning. However, most of them are closed-source: Marscode Agent (Liu et al., 2024), Composio SWE-Kit (Composio, 2024), CodeR (CodeR, 2024), Lingma (Ma et al., 2024), Amazon Q (Amazon, 2024), IBM Research SWE-1.0 (IBM, 2024), devlo (devlo, 2024), Gru (gru, 2024), and Globant Code Fixer Agent (Globant, 2024). Here, we focus on the open-source approaches.

A notable early method is SWE-Agent (Yang et al., 2024a), which has only localization and generation and leverages an LLM planner to drive the patching process. To assist the planner in calling functions within each component, SWE-Agent provides an Agent-Computer Interface (ACI), which grants LLMs the ability to execute bash commands and handle file operations (e.g., `file_open` and `func_edit`). Follow-up works improve SWE-Agent by either improving its current components (AutoCodeRover (Zhang et al., 2024b)) or incorporating additional components (Moatless (moatless, 2024; Antoniades et al., 2024) and SpecRover (Ruan et al., 2024)). Notably, Moatless and SpecRover add a validation component. This component first lets LLM generate an input that can trigger the issue (denoted as "Proof-of-Concept (PoC)") and then runs the PoC against the generated patches to decide if they fix the issue.

So far, the SOTA open-source tool in this category is Open-Hands (Wang et al., 2024b), which is inspired by the Code-Act Agent (Wang et al., 2024a). OpenHands has three com-

ponents: localization, generation, validation. Its validation follows a similar idea as SpecRover, i.e., reproducing and executing PoC to decide if the issue is fixed. Similar to the SWE-agent, OpenHands also designs an ACI for the agent.

Limitations. Agent-based planning approaches inherently suffer from two critical limitations. First, as probabilistic models, LLMs intrinsically have randomness. The randomness is aggregated and amplified when the model is making all critical decisions during the patching. This will significantly jeopardize the stability and reliability of the patching agents, hindering their real-world usage. Second, to reduce randomness, existing approaches conduct multiple samples and trials, and ensemble them to obtain the LLMs' decisions. Moreover, LLMs often need multiple trials to obtain a correct decision. All these extra samples and trials significantly raise computational costs as well as financial costs as they need to use commercial models.

**Rule-based planning.** Agentless (Xia et al., 2024) is the SOTA method following rule-based planning. Agentless strictly follows a pre-defined sequential workflow, comprising localization, generation, and validation. Specifically, for localization, Agentless designs a three-step procedure (file, function, line), where LLM is used to pinpoint the root cause at each step. It directly queries LLM without leveraging the rich information in the code structure. Agentless's generation feeds the root cause and issue description to LLM and lets the model generate patch candidates. It simply stacks the input information together without using advanced prompting strategies. Its validation is similar to the agent-based planning methods introduced above. RepoGraph (Ouyang et al., 2024) improves the localization by providing a repository-level graph but without changing other components. Having a pre-specified workflow makes these methods more stable than agent-based planning methods. It also allows the agent to integrate human knowledge.

Limitations. Agentless's sequential workflow is overly restrictive. The agent cannot refine the root cause, generated patches, and PoCs if the patch candidates cannot pass the validation. It has to start over again, which is less efficient. In addition, as discussed above, the individual components of Agentless and RepoGraph also have flaws.

## 3. Methodology of PatchPilot

### 3.1. Technical Overview

**Problem definition.** Given a buggy code repository written in `Python`, denoted as $\mathcal{R}$, which contains a set of functionalities $\mathcal{F} = f_1, f_2, \ldots, f_n$ written in different files. The repository may have one or more issues, where each issue $\beta_i$ has an issue description written in text, denoted as $D_i$. The issue $\beta_i$ affects a subset of functionalities, denoted as $\mathcal{F}_{B_i} \subseteq \mathcal{F}$. A successful patch, denoted as $p$, should fix

all functionalities in $\mathcal{F}_{B_i}$ while preserving the behaviors of the unaffected functionalities $\mathcal{F}s = \mathcal{F} \setminus \mathcal{F}_{B_i}$. Our main goals are twofold. First, we aim to resolve as many issues as possible across different issues and diverse repositories. Second, we also aim to maximize the stability and reduce the cost of our patching framework. We believe *stability and cost-efficiency* are critical for real-world applications of a patching tool. An unstable tool that produces only one correct patch across multiple runs significantly hinders its applicability for critical bugs. Furthermore, if the tool is too costly to use, it limits its usage by ordinary users.

**Rationale behind PatchPilot.** Recall from Section 2 that we discussed the advantages and disadvantages of rule-based versus agent-based planning. In general, agent-based planning is more expensive and less stable than rule-based planning. However, it may give a higher optimal issue resolved rate than rule-based planning, as the LLM planner can explore more tailored workflows for different issues. In contrast, rule-based planning relies on a uniform workflow across different scenarios, which may not be effective in certain instances. As such, if the primary goal is to maximize the resolved rate on certain benchmarks, agent-based planning should be the preferred strategy. Indeed, most existing tools follow this approach, especially in industry settings with much more resources than academia. However, as mentioned above, a high resolved rate is not our sole goal, nor is it the only metric for evaluating a good patching agent. Stability and cost-efficiency are equally important as the resolved rate given that we are developing a tool that can be used in real work rather than just exploring the boundary of LLM agents. As such, we choose to follow rule-based planning in our patching agent.

Figure 1 illustrates the workflow of PatchPilot. It consists of five phases: reproduction, localization, patch generation, validation, and patch refinement. As discussed in Section 2, localization and generation are commonly included in existing approaches. We add three additional components to improve the overall patching effectiveness and efficiency. The reproduction and validation components are crucial for determining patch quality and selecting the correct patch candidates for deployment. Some advanced patching agents also include these components; in Section 3.2, we will specify how we designed ours to be more accurate and stable. Refinement is a unique component in PatchPilot, as we observe that improving a partially correct patch based on validation feedback is often more effective and efficient than generating a new patch from scratch. This aligns with human experience, as a correct patch often requires multiple rounds of testing and refinement.

**Workflow of PatchPilot.** As shown in Figure 1, given the input of the codebase $\mathcal{R}$ and the description $D_i$ of the target issue $\beta_i$, PatchPilot first calls reproduction to recover

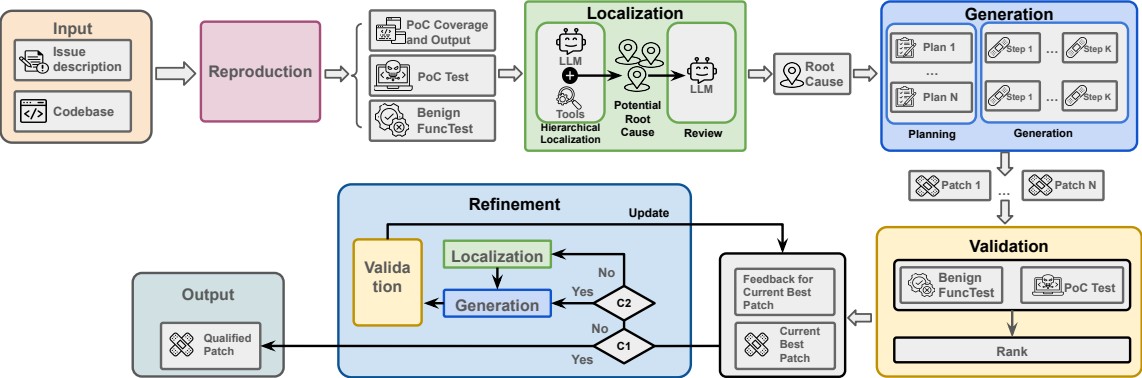

*Figure 1.* Overview of PatchPilot. The system processes input through reproduction, localization, generation, validation, and refinement to obtain a final patch. Both localization and generation have two phases. The validation considers both PoC and functionality tests. Finally, the iterative refinement involves two conditions: C1 checks if the patch passes all tests, if yes, the patch will be outputted; if no, C2 then checks if the current patch passes a new test compared to the previous round.

a set of testing cases, including PoC (a test that can trigger the issue) and benign functionality tests. PatchPilot runs the PoC and obtains the files it covered and the outputs. Then, the localization component takes as input $\mathcal{R}$, $D_i$, and information related to the PoC and outputs the root cause (specific lines causing the issue). Similarly to Agentless (Xia et al., 2024), our localization also follows a hierarchical workflow but with additional tools to better extract and leverage the program structures. After identifying the root cause, the generation component generates $N$ patch candidates at once. As discussed in Section 3.4, the key novelty here is separating planning and generation and leveraging multiple prompting strategies to encourage patch diversity. The generated patch candidates are then fed to the validation component, which ranks the candidates based on their results of running the PoC and functionality tests. If the validation cannot find a qualified patch that passes all available tests, the refinement component will be called to refine the top-ranked patch candidate or refine the localization based on the validation results. PatchPilot iteratively performs refinement and validation until it either identifies a qualified patch or reaches the maximum allowed number of generated patches ($N_{\max}$).

### 3.2. Reproduction and validation

**Reproduction.** We introduce three improvements over existing work (Xia et al., 2024). First, reproduction in existing patching agents directly provides an LLM with $\mathcal{R}$ and $D_i$ and prompts it to generate a PoC. However, $D_i$ often includes only short code snippets related to the issue without specifying necessary dependencies and configurations (e.g., the issue descriptions of Django typically do not have environment setups). Without such information, the generated PoCs often fail to run successfully. To address this challenge, we propose a *self-reflection-based PoC reproduction*,

which is similar to the Reflexion mechanism designed for language agents (Shinn et al., 2024). During the process, we let LLM iteratively generate and refine the generated PoC for certain iterations. We carefully construct our prompts to guide the LLM focus on checking and correcting 1) whether any key dependencies and configurations are missing; and 2) whether the PoC actually reproduces the target issue. If the reproduction fails to generate a valid PoC within the maximum iterations, we proceed without a PoC. Second, different from existing works that only use the generated PoCs, we extract a more complete set of information based on PoCs. This includes files covered by running the PoC, stack traces and outputs. As we will discuss later, this extra information helps localization and refinement. Third, we utilize LLM to identify three functionality test files from $\mathcal{R}$ that are most relevant to the target issue (each file may contain multiple testing cases). These functionality tests enable the validation component to decide if the patch candidates preserve the functionalities of $\mathcal{F}_S$, an important metric for a successful patch. More details about additional information retrieved based on PoCs are discussed in Appendix A.

**Validation.** The simple validation strategy utilized in existing works (Tao et al., 2024; Globant, 2024; Ma et al., 2024) is just to feed the patch candidates and the related information to an LLM and let it select the most qualified one. A more advanced strategy (Liu et al., 2024; Wang et al., 2024b; Arora et al., 2024) is to run the generated PoC and let an LLM decide whether the patches fix the issue based on their outputs. As mentioned above, ensuring the correctness of the original functionalities is as important as fixing the issue. As such, we include the functionality tests recovered by our reproduction in the validation. Specifically, we first run our PoC on the patch candidates and use an LLM as a judge for its evaluation. Since no assertions are available for bug fixing, this serves as the only feasible solution. We then

also run the functionality tests and decide whether they pass based on their given assertions. Finally, we rank the patches based on the tests they pass. As specified in Appendix A, we prioritize patches that pass PoC tests over functionality tests during ranking.

## 3.3. Localization

**Key challenges.** Some existing localization directly query an LLM to identify the root cause at a line level (Yang et al., 2024a; Arora et al., 2024; Zhang et al., 2024b). Although they provide the LLM with tools to retrieve information from the codebase and allow it to refine its results, it is still difficult for LLMs to directly perform localizations at the line level. Besides, most agent-based tools incur high costs because they need to maintain the LLM agent's context history during localization. Agentless designs a hierarchical workflow, which first identifies the issue-related files, then the functions, and lastly the lines. This method gradually zooms into and makes the task easier at the line level as it filters out the majority of the non-related functions in the earlier steps. At each step, Agentless lets the LLM make decisions only based on the issue description. This approach has three critical limitations. First, the information in issue descriptions is diverse and not all of them have useful information for localization. For example, some descriptions only specify error messages and PoC-related information that is not helpful for localization. Second, this method lacks a direct mechanism for retrieving details directly from the codebase. Third, in most cases, the localization returns only the root cause it is confident about as a few lines of code. While this information is accurate, it is often insufficient for writing a correct patch due to the lack of necessary context.

**Our design.** We follow the three-step procedure in Agentless given it is more stable and efficient than letting LLM directly do line level localization. First, to address the limitation of inconsistent issue descriptions, we provide the LLM with the PoC code and information after running it (i.e., files it covered, stack trace, and running outputs). This enables the LLM to access more comprehensive information, such as key functions or classes invoked in the PoC and the stack trace, which is particularly useful for cases where only the code to reproduce the issue is provided in the issue description. For example, the files covered by PoC can help filter out some files irrelevant to the target issue, reducing the search space, especially for codebases with many files. Second, to enable the LLM to extract and leverage more information from the codebase, we add a set of tools to the localization component. These tools allow the LLM to search for class definitions and function definitions, or perform fuzzy string matching to locate and return relevant files. These tools provide precise search capabilities and can handle both class/function level information and line level

details. Appendix A has more details on the tools we integrate. Third, as shown in Figure 1, we add a review step that lets an LLM retrieve code snippets related to the current root cause. As mentioned above, localization oftentimes returns overly precise root causes that fail to include necessary context or even do not fully cover all root causes. Identifying more contexts is important to generate correct and complete patches. Note that we still constrain the maximum length of the final root cause to make sure not to overwhelm the generation with excessive context.

## 3.4. Patch Generation

**Key challenge.** Most existing patch generation components simply stack the related information and feed them to LLM for patch generation. Such a simple solution has two critical challenges. First, LLMs typically give incomplete patches. This is because fixing an issue often requires modifications across multiple locations or involves multiple steps, making it difficult to generate a complete patch in one shot. In addition, the incomplete root causes also lead to this issue. Second, being able to generate diverse patches is also crucial to increasing the likelihood of finding a successful patch within certain trials. Moreover, we find that simply increasing the temperature still results in similar patches. We need other strategies to increase patch diversity, enabling the agent to search for more potential solutions.

**Our design.** First, as shown in Figure 1, rather than directly generating the patch, PatchPilot breaks down the generation process into planning and generation. The planning phase first queries the LLM to generate a patch plan with multiple steps. The generation phase then generates the patch following the plan. After finishing each step in the plan, we also include a lightweight in-generation validation with lint and syntax checks, and reconduct this step if the check fails. This design is motivated by the Chain-of-thoughts prompting strategy (Wei et al., 2022). That is, having a plan explicitly forces the LLM to break down the patch generation into multiple steps. This helps the model to better reason about the patch task, encouraging it to provide more complete patches. Besides the in-generation validation can identify and fix errors at an early stage, improving the patch efficiency. Second, to enhance the diversity of the generated patch candidates, we design three types of prompts for plan generation. These prompts explicitly guide the LLM to produce patching plans with different focuses: a comprehensive and extensive patch designed to prevent similar issues, a minimal patch with the smallest possible modifications, or a standard patch without any specific instructions. Appendix D contains more details on the prompts that we use. As demonstrated Figure 1, we will generate $N$ plans following the pre-specified prompts and thus produce $N$ patch candidates in each batch.

*Table 1.* Comparison of PatchPilot and five baselines on the two benchmarks. "Agent-based" and "Rule-based" refer to agent-based planning and rule-based planning, respectively. "-" means not available. Note that Globant does not report the results on SWE-bench Verified, and CodeStory does not report their result on SWE-bench Lite. They both do not disclose the LLM model(s) in their agents.

| | Patching agent | Open-source | SWE-bench Lite | | | SWE-bench Verified | | |
|---|---|---|---|---|---|---|---|---|
| | | | LLM | Resolved% | Cost ($) | LLM | Resolved% | Cost ($) |
| **Agent-based** | AutoCodeRover | ✓ | GPT-4o | 30.67% (92) | 0.65 | Claude-3.5-Sonnet | 51.80% (259) | 4.50 |
| | OpenHands | ✓ | Claude-3.5-Sonnet | 41.67% (126) | 1.33 | Claude-3.5-Sonnet | 53.00% (265) | 0.78 |
| | Globant | ✗ | - | 48.33% (145) | 1.00 | - | - | - |
| | CodeStory | ✗ | - | - | - | - | 62.20% (311) | 20.00 |
| **Rule-based** | Agentless | ✓ | Claude-3.5-Sonnet | 40.67% (123) | 1.12 | Claude-3.5-Sonnet | 50.80% (254) | 1.19 |
| | PatchPilot | ✓ | Claude-3.5-Sonnet | 45.33% (136) | 0.97 | Claude-3.5-Sonnet | 53.60% (268) | 0.99 |

## 3.5. Patch Refinement

Recall that refinement is a unique component in PatchPilot that existing works do not have. The motivation for adding this component is to better leverage the validation feedback and the current partially correct patches. As shown in Section 4.3, refining existing parties based on validation results is more effective and efficient than re-generating patches from scratch. More specifically, as demonstrated in Figure 1, PatchPilot focuses on refining the top-ranked patch in the current batch. It feeds the current batch and its validation result back to the generation component and asks it to generate a new batch of patches. The generation still follows the planning and generation workflow. Here, when generating the plans, we design the prompt to guide the model to correct the failed testing cases of the current patch. This process continues until a qualified patch that passes all validations is generated, or the total number of generated patches reaches the predefined limit of $N_{max}$. Note that if the patches generated in a whole batch do not pass any new tests, we rerun the localization with the validation results to obtain a new root cause. This additional step gives PatchPilot the opportunity to leverage information from later components to correct localization errors and ultimately succeed in generating qualified patches.

## 4. Evaluation

We evaluate PatchPilot from the following aspects: First, we perform a large-scale comparison of PatchPilot with both SOTA open-source and closed-source methods on the SWE-bench Lite and SWE-bench Verified patching benchmark (Jimenez et al., 2023), showcasing PatchPilot's ability to balance patching accuracy and cost-efficiency. Second, we conduct a stability analysis on PatchPilot and Open-Hands, demonstrating PatchPilot's rule-based planning is more stable than the SOTA agent-based planning. Third, we conduct an ablation study to quantify the contribution of each component to PatchPilot's overall performance. Finally, we show PatchPilot's compatibility and performance on different models, including GPT-4o (OpenAI, 2024a), Claude-3.5-Sonnet (Anthropic, 2023), and a reasoning model o3-mini (OpenAI, 2024b). We failed to integrate DeepSeek-r1 (DeepSeek, 2025) due to the problems with their APIs (See Appendix C.3).

### 4.1. PatchPilot vs. Baselines on SWE-bench

**Setup and design.** We utilize the *SWE-bench* (Jimenez et al., 2023) benchmark, where each instance corresponds to an issue in a GitHub repository written in `Python`. Specifically, we consider two subsets: *SWE-bench Lite* (SWE-Bench, 2023a), consisting of 300 instances, and *SWE-bench Verified* (SWE-Bench, 2023b), comprising 500 instances that have been verified by humans to be resolvable.

We mainly compare PatchPilot with three SOTA open-source methods: two agent-based planning methods Open-Hands (Wang et al., 2024b) and AutoCodeRover (Zhang et al., 2024b), and a rule-based planning method Agentless (Xia et al., 2024). We also compare it with two closed-source methods that have cost reported: Globant Code Fixer Agent (Globant, 2024) (Globant for short) and CodeStory Midwit Agent (CodeStory, 2024) (CodeStory for short). In Appendix C.1, we include a more comprehensive comparison of PatchPilot against 29 other tools, showing our positions on the SWE-bench leaderboard. Given our goal of addressing stability and cost together with the resolved rate, comparing closed-source methods that have a higher resolved rate but without cost is not our focus. Most of these methods follow agent-based planning that may cost way more than ours. For example, CodeStory mentions that it costs them $10,000 to achieve 62.2% on the SWE-bench Verified benchmark (SWE-Bench, 2023b), whereas PatchPilot achieves a 53.60% with less than $500 (20× cheaper). In addition, as shown in Section 4.2, agent-based planning is less stable than rule-based planning.

To align with most methods, we use the Claude-3.5-Sonnet model as the LLM in PatchPilot. Appendix B shows our implementation details. We report two metrics *Resolved Rate* (%): the percentage of resolved instances in the benchmark,[1] and *Average Cost* ($): the average model API cost

---

[1] An instance/issue is resolved means the patch fixes the issue while passing all hidden functionality tests.

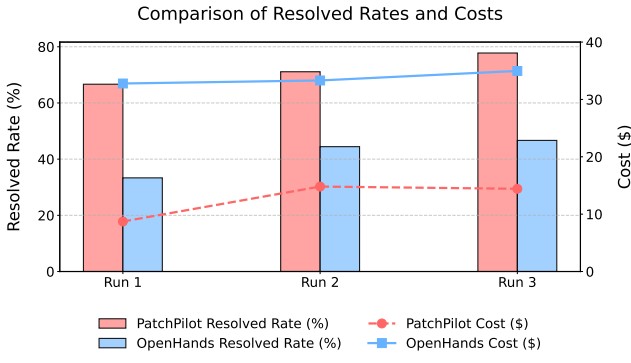

*Figure 2.* PatchPilot vs. OpenHands in the resolved rate (bars) and the total cost (lines) on 45 instances from SWE-bench Lite.

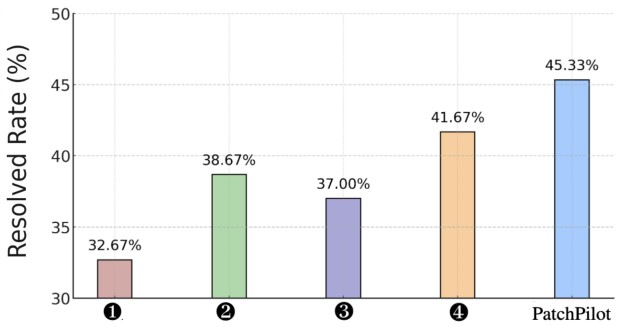

*Figure 3.* Ablation study results on the SWE-bench Lite benchmark. ❶∼❹ refers to *Base Local+Gen*, *Our Local+Gen*, *Our Local+Gen+PoC*, and *Our Local+Gen+Val*, respectively.

of running the tool on each instance. For the baselines, we retrieve their performance from their submission logs on the SWE-bench and their papers and official blogs.

**Results.** Table 1 shows the performance of PatchPilot and selected baselines on two subsets of the SWE-bench benchmark. Although, on both benchmarks, the closed-source methods achieve the highest performance, their internal design and methodology are not publicly available and we cannot assess their stability. Notably, the cost of CodeStory is 20× higher than PatchPilot. The cost of Globant is more comparable to PatchPilot on SWE-bench Lite, but we cannot assess their performance and cost on SWE-bench Verified. Among open-source methods, OpenHands achieves higher resolved rates than the rule-based planning tool, Agentless, on both benchmarks. However, OpenHands has a higher cost than Agentless on SWE-bench Lite when using the same Claude-3.5-Sonnet. This result validates our discussion in Section 3.1, rule-based planning is more cost-efficient than agent-based planning, and agent-based planning has the potential to achieve higher optimal resolved rates.

In comparison, PatchPilot demonstrates a clear advantage in balancing resolved rate and cost. On SWE-bench Lite, it resolves 45.33% (136/300) of the issues, outperforming all open-source methods with a low cost of $0.97 per instance. Similarly, on SWE-bench Verified, PatchPilot achieves a resolved rate of 53.60% (268/500), surpassing all open-source methods while maintaining the same cost efficiency of $0.99 per instance. These results highlight the efficacy and cost-efficiency of PatchPilot.

### 4.2. PatchPilot vs OpenHands in Stability

**Setup and design.** We compare the stability of PatchPilot and OpenHands, the SOTA open-source agent-based planning tool. We find 102 common instances resolved by PatchPilot and OpenHands in the SWE-bench Lite benchmark and randomly select a subset of 45. We run PatchPilot

and OpenHands on these instances three times with GPT-4o model and different `Python` random seeds. We report and compare their resolved rate and total cost in each run.

**Results.** Figure 2 shows the resolved rate and costs of PatchPilot and OpenHands across three runs. As shown in the figure, PatchPilot consistently resolved more instances, achieving 30, 32, and 35 resolved instances in the three runs, with a standard deviation of 2.52. In comparison, OpenHands resolved only 15, 20, and 21 instances, with a higher standard deviation of 3.21. The lower standard deviation of PatchPilot demonstrates its stability, which further validates our discussion about rule-based planning vs. agent-based planning in Section 3.1. Additionally, Patch-Pilot demonstrated a clear advantage in terms of cost efficiency, with costs of $8.72, $14.81, and $14.42 for the three runs, resulting in an average of $12.65 per run. This is substantially lower than OpenHands, which incurred costs of $32.78, $33.31, and $34.97, with an average of $33.69 per run. These results further highlight PatchPilot's ability to achieve higher resolved rates with greater stability and at a lower cost.

### 4.3. Ablation Studies

**Setup and design.** We conduct a detailed ablation study to investigate the efficacy of key designs in PatchPilot. We use the full SWE-bench Lite benchmark and the Claude-3.5-Sonnet model for all variations of our method. Specifically, we consider the following four variations: ❶*Base Local+Gen*: We combine simple localization without providing the LLM with tools or a review step, along with simple generation without the two-phase design (Figure 1). We choose the final patch by majority voting. ❷*Our Local+Gen*: We combine PatchPilot's localization and generation components together with majority voting for final patch selection. Comparing ❶ with ❷ can assess the effectiveness of our proposed techniques for localization and generation. ❸*Our Local+Gen+PoC*: We further add our validation component

to ❷ but with only the PoC tests (the validation strategy employed by most existing tools). Comparing ❷ with ❸ can assess the effectiveness of having PoC validation instead of simple majority voting. ❹*Our Local+Gen+Val*: We add the full validation component, comparing ❸ with ❹ can assess the efficacy of having functionality tests in validation. Finally, comparing ❹ with PatchPilot can assess the importance of having an additional refinement component.

**Results.** Figure 3 shows the resolved rates across different variations and our final method. By incrementally building upon the core functionalities of PatchPilot, we evaluate the contributions of individual components to the overall patching performance.

Localization and generation. First, we can observe that ❶ with the simple localization and generation only get a resolved rate of 32.7% (98/300). In contrast, ❷ with our improved localization and generation increases the resolved rate to 38.7% (116/300). This result first confirms the challenges of simple localization and generation designs discussed in Section 3.3 and Section 3.4, as they prevent ❶ from achieving a better performance. More importantly, it validates the effectiveness of our designs in adding tools and a review step in localization and the two-step procedure (i.e., planning and generation) in the generation.

PoC and functionality validation. ❸ with our localization and generation as well as PoC validation unexpectedly lowers the resolved rate to 37.00% (111/300). This result suggests that relying solely on PoC validation may resolve the targeted issue while introducing new functional issues. As such, when functionality tests are added, ❹ significantly improves the resolved rate to 41.67% (125/300). This result shows that functionality tests play a crucial role in identifying and filtering out the patches that fix the target issues but break the original functionalities of the codebase. As mentioned above, a patch must pass all hidden functionality tests to be marked as a success; having functionality tests is important to filter out false positives.

Refinement. Finally, adding our refinement component on top of ❹ improves the resolved rate from 41.67% to 45.33%. The result demonstrates the effectiveness of our refinement design. It also justifies our claim in Section 3.5 that generating new patches from scratch when the current trial fails is less effective than refining the partially correct patches based on the validation feedback.

### 4.4. PatchPilot on Different Models

**Setup and design.** To demonstrate the compatibility of PatchPilot to different LLMs, we conduct an experiment that integrates PatchPilot with three SOTA LLMs: two general models GPT-4o and Claude-3.5-Sonnet, and one reasoning model: o3-mini. We select a subset of 100 instances from

the SWE-bench Lite benchmark; all these 100 instances have been successfully resolved by at least one method ranked Top-10 on the SWE-bench leaderboard. We run PatchPilot with the selected models on these instances and report the final resolved rate. We keep all other components the same and only change the model to show the impacts of the different models.

**Results.** The resolved rate of PatchPilot with different models are: GPT-4o: 19.00%; Claude-3.5-Sonnet: 39.00%, and o3-mini: 43.00%. o3-mini achieves the highest resolved rate, indicating having inference-phase reasoning capabilities is helpful not only for general math and coding tasks but also for the specialized patching task. Note that although we cannot directly compare with the results reported from official reports (anthropic, 2024; OpenAI, 2024c; Under, 2024), as they conduct their testing on the SWE-bench Verified benchmark. However, they follow the same trend: o3-mini > Claude-3.5-Sonnet > GPT-4o. It is also worth noting that PatchPilot with Claude-3.5-Sonnet on the SWE-bench Verified benchmark reports a higher resolved rate than the official report from Claude-3.5-Sonnet and OpenAI-O1 model. Although the full o3 reports a resolved rate of 71.7%, it do not disclose any details about the system design, cost, and stability. Overall, this experiment demonstrates the compatibility of PatchPilot to different models as well as the efficacy of having a reasoning model in PatchPilot.

## 5. Additional Formal Verification Component

To strengthen the reliability of our automatically generated patches, we introduce a formal-verification stage built on CrossHair (Schanely, 2022). CrossHair takes as input a function together with a logical contract, consisting of pre-conditions and post-conditions. Here, the pre-conditions capture assumptions that must hold when the function is invoked, and the post-conditions describe the properties expected to hold when the function returns. CrossHair works by symbolically executing the function under the given pre-conditions and using the Z3 SMT solver to check whether the post-condition holds for all valid inputs. If an input is found that violates the post-condition, CrossHair reports it as a counter-example.

In our workflow, the verification stage for each candidate patch includes the following three steps:

**Contract Generation.** We prompt the LLM with the issue description and the original versions of the functions modified by the patch. For each function, we ask the LLM to generate type annotations for the function signature and a logical contract containing pre-conditions and post-conditions.

**Contract Validation.** We perform an initial validation step to filter out clearly incorrect logical contracts generated by the LLM. Specifically, for each function modified by the

patch, we insert the generated type annotations and logical contract into the original version of the function and run CrossHair. If CrossHair identifies at least one counter-example triggering an `AssertionError` in any of the original functions, the set of contracts for the patch is considered plausible. If no counter-example is found for any function, we regenerate the logical contracts and type annotations for all functions, repeating this process up to a maximum of $N_{regen}$ iterations. If plausible contracts still cannot be generated, we prompt the LLM again with the issue description and original functions, and allow it to remove branches and logic in the original functions that are irrelevant to the post-conditions, and keep only the key logic. If plausible contracts still cannot be generated after this step, we mark the instance as unverified.

**Formal Verification.** We insert the logical contracts and type annotations into each patched function and run CrossHair. If CrossHair cannot find any counter-example that satisfies the pre-conditions but violates the post-conditions for any patched function in the instance, we consider the instance verified.

We evaluate this procedure on the SWE-bench Lite with the o3 model with with $N_{regen} = 3$, and formally verify eleven patches. This modest number is due to the current limitations of Python-based formal verifiers, which struggle with intricate data structures and deeply nested or dynamic function calls.

## 6. Discussion

**Resolved rates vs. stability and cost.** As a rule-based planning method, PatchPilot achieves a well-balanced trade-off between resolved rates, stability, and cost-efficiency. We believe that emphasizing stability and cost-efficiency is essential for a patching agent to be practical in real-world applications. Although PatchPilot is not on the top of the leaderboard, it has been shown to be a stable and affordable method, confirming its practicality. Furthermore, PatchPilot outperforms all open-source methods on the leaderboard, establishing reasonably good performance.

**Static analysis vs. LLMs.** We tried multiple static program analysis approaches in different components which were not effective and cannot outperform LLMs. First, we added function summaries for the functions in the root cause to the generation component. It improves the performance of PatchPilot with GPT-4o. However, it is not helpful when using more advanced models (o3-mini and Claude-3.5-Sonnet), indicating that advanced models may be able to infer function behaviors. In validation, we tried to apply rule-based criteria to the PoC outputs to decide whether an issue is fixed. This is worse than using LLM as a judge, given that many issues are "logical bugs" that do

not cause crashes. LLMs can better understand the issue and make decisions based on PoC outputs. We also tried to use CodeQL to infer patch-related locations that needed to be changed together with the current patch, but it failed due to CodeQL's limited performance.

**Complex prompting strategy.** We tried Tree of Thoughts (ToT) (Yao et al., 2024) in generation, i.e., generating multiple candidates for each step of a plan. While significantly increasing costs, this approach does not improve the resolved rate given the LLMs cannot generate candidates with enough diversity for specific patching steps.

## 7. Conclusion and Future Works

We present PatchPilot, a stable and cost-efficient patching agent driven by a rule-based planning workflow. We design five important components and each element has its own customized designs. Our experiment demonstrates the effectiveness of PatchPilot and its individual components. This work points to a few promising directions for future work. First, we can explore combining agent-based planning and rule-based planning, which potentially can reach a higher resolved rate with reasonable stability and cost-efficiency. Second, as discussed in Appendix C.4, a number of failure instances are caused by the fact that LLMs tend to give overly simple patches. It is worth investigating fine-tuning specialized patching models from SOTA general LLMs which can better understand and reason about the issues and give more comprehensive patches.

## Acknowledgements

This work was supported in part by ARL Grant W911NF-23-2-0137. We gratefully acknowledge the support of FAR AI, OpenAI, and Berkeley RDI.

## Impact Statement

PatchPilot advances automated software patching by introducing a stable and cost-efficient framework that balances performance with real-world practicality. While our method may achieve lower resolved rates compared to some closed-source solutions, we believe this tradeoff is justified by significantly reduced costs and improved stability - critical factors for widespread adoption in production environments. Our rule-based planning approach helps prevent the unpredictability and high costs associated with fully agent-based systems while maintaining competitive patch generation capabilities. However, we acknowledge that further research is needed to address the complex program logic understanding challenges. The framework's open-source nature and emphasis on practical deployment considerations contribute to making automated patching more accessible and reliable

for real-world software maintenance.

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

# A. Additional Technical Details

**Tools provided in localization.** For file-level localization, we provide LLM with three tools, which are *search_func_def*, *search_class_def*, and *search_string*. *search_func_def* and *search_class_def* take the function or class name as input and output the file containing the corresponding function or class definition based on exact matching. *search_string* takes a string as input and returns the file that contains this string the most times. If no file is found, we perform a fuzzy match with a decreasing similarity threshold until a match is found or timeout. This is because the string searched by the LLM, often an error message in $D_i$, is typically a formatted string in the code and may not exactly match the one in $D_i$. We chose to provide these three tools because they could cover most of the information provided in the issue descriptions.

Note that the use of these tools is predefined and incorporated into the file-level prompt at the start of the localization process, rather than being determined dynamically by the LLM. This ensures that our approach remains firmly rooted in rule-based planning.

**Additional PoC information.** We also attempt to identify the issue-introducing commit by finding the first historical commit where the PoC triggers the issue and its preceding commit. This is achieved by using a binary search to find the latest commit whose PoC output matches the current commit. Then, we prompt LLM to analyze the PoC output to determine if the preceding commit produces expected results or triggers unrelated errors. If it is the former case, we consider we have found the issue-introducing commit. We extract the code difference between the issue-introducing commit and its previous commit. This code difference provides valuable insights into how the bug was introduced, which is highly beneficial for both localization and generation. In Section 4.1, PatchPilot successfully identified the issue-introducing commit in 18 out of 300 instances on the SWE-Bench Lite benchmark, among which 13 were resolved, achieving a resolved rate of 72.22%. In the SWE-Bench Verified benchmark, the issue-introducing commit was found in 39 instances, with 24 successfully resolved, achieving a resolved rate of 61.53%. These high resolved rates further demonstrate the effectiveness of the issue-introducing commit.

**Ranking criteria.** We design our ranking criteria to prioritize the PoC test over benign functionality tests. The rank of a patch $p$ is defined as

$$RK_p = \mathbb{1}(\text{PoC\_failed}) + \frac{num_{\text{failed\_func\_test}}}{num_{\text{executed\_func\_test}}}, \tag{1}$$

where $RK_p$ is the rank of the patch $p$, $\mathbb{1}(\text{PoC\_failed})$ is an indicator function determined by whether the PoC fails, $num_{\text{failed\_func\_test}}$ is the number of failed functionality tests, and $num_{\text{executed\_func\_test}}$ is the number of executed functionality tests. We rank the patches based on the reverse order of $RK_p$ (i.e., the lower the $RK_p$, the higher the ranking). If multiple patches have the same rank, we leverage an LLM to determine which one is the best.

# B. Implementation

**Reproduction.** For reproduction, we set the iteration limit as 7 for the PoC generation. We generate at most one PoC for each instance. If a PoC is successfully obtained, we leverage the Python coverage package to get the files that are covered during the execution of the PoC. We provide LLM with $\mathcal{R}$'s directory tree structure and the issue description $D_i$, directly prompting LLM to retrieve three existing test files as benign functionality tests.

**Localization.** In localization, we perform a hierarchical workflow, including file level, class and function level and line level localization, followed by a review step. At the file level, PatchPilot constructs a tree-structured repository representation, filtered based on the PoC coverage, retaining only the files executed during PoC execution. The issue description, the repository representation, and the set of tools described in Appendix A are then provided to the LLM, prompting it to return the top 5 files most relevant to the issue. To achieve self-consistency (Ahmed & Devanbu, 2023), we perform file-level localization 4 times and perform majority voting to get the top 5 files. These files are inputs of the class and function-level localization. At the class and function level, the LLM is provided with the signatures and comments of classes and functions extracted from the retrieved files and is prompted to identify functions and classes likely related to the issue. The number of functions and classes returned by LLM is not limited. At the line level, we provide the complete source code of the identified classes and functions to the LLM and prompt it to localize to specific lines. The class and function-level localization and the line-level localization are both performed 4 times, with the results of each step merged. The localized lines, along with the surrounding code within a ±15-line range, are considered the root cause that will be provided to generation. At the end, we prompt LLM to perform review. If the root cause is less than 150 lines of code, we prompt LLM to retrieve more code

*Table 2.* SWE-Bench Verified leaderboard excerpts from tools with more than 50% resolved rate

| Tool | LLM | %Resolved |
|---|---|---|
| Blackbox Ai Agent | NA | 314(62.80%) |
| CodeStory Midwit Agent + swe-search | NA | 311(62.20%) |
| Learn-by-interact | NA | 301(60.20%) |
| devlo | NA | 291(58.20%) |
| Emergent E1 | NA | 286(57.20%) |
| Gru | Claude-3.5-Sonnet | 285(57.00%) |
| EPAM AI/Run Developer Agent | NA | 277(55.40%) |
| Amazon Q Developer Agent | NA | 275(55.00%) |
| PatchPilot | Claude-3.5-Sonnet | 268 (53.60%) |
| OpenHands + CodeAct v2.1 e | Claude-3.5-Sonnet | 265(53.00%) |
| Google Jules | Gemini 2.0 F | 261(52.20%) |
| Engine Labs | NA | 259(51.80%) |
| Agentless | Claude-3.5-Sonnet | 104(50.80%) |
| Solver | NA | 100(50.00%) |
| Bytedance MarsCode Agent | NA | 100(50.00%) |

snippets related to the root cause. Otherwise, we prompt LLM to check whether the current root cause is correct and fix it if LLM determines that it is incorrect.

**Generation.** For the experiments in Section 4, we set $N_{\max}$ as 12, and $N$ as 4. Thus, PatchPilot generates 4 plans and 4 patches in a single batch and can generate up to 12 patches for one instance. We constrain the maximum number of steps in a plan to 3. For the first batch, we use the prompts as specified in Appendix D, generating one patch for the comprehensive-patch prompt, one for the minimal-patch prompt and two patches for the standard-patch prompt. For the following batches, we only use the standard-patch prompt. This is done to facilitate caching and reduce costs. Following previous work (Aider, 2024), we prompt LLM to generate patch in Search/Replace edit format and transfer them to git diff format by ourselves. For each edit generated for a step of a plan, we utilize Python's ast library for syntax check and Flake8 for lint check. If LLM cannot generate an edit that passes both the syntax check and the lint check, we skip generation for the current step.

**Validation.** For the PoC test, we run the PoC, collect the outputs and provide it to LLM, along with $D_i$ and the PoC code, and prompt LLM to judge whether the issue is fixed. For functionality tests, we implement a parser that extracts the function names of the failed test from the test output. We assign each patch a rank as specified in Appendix A.

**Refinement.** For the refinement, if the current batch of patch pass no new functionality tests, we rerun localization in the next batch. For the file level localization, we force the LLM to return one file that is not within the previous localized files. For the function and class level localization and the line level localization, we force LLM to return code that is not within the previous localization results. If there is still no patch that passes any new test in the next batch, we discard the newly localized results and restore the original localization results before rerunning localization in the following batches.

## C. Additional Experiments and Results

### C.1. More Comparison on SWE-Bench.

Table 3 and Table 2 are about the current results on the SWE-Bench Lite/Verified, which show that PatchPilot has the highest performance among open-source tools and is competitive with closed-source tools in terms of resolved rate and cost.

### C.2. More Analysis on Stablility Test.

Next, we go deep to analyze the resolved instances in each run in Section 4.2. Figure 4 provides a Venn diagram comparison of the inter-

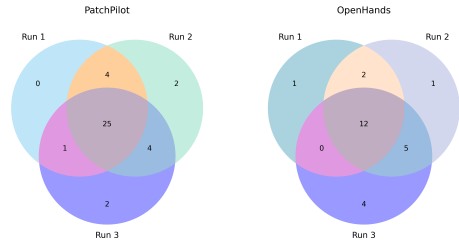

*Figure 4.* Venn of resolved instances in rounds

*Table 3.* SWE-Bench Lite leaderboard excerpts

| Tool | LLM | %Resolved | $Avg. Cost |
|---|---|---|---|
| Blackbox Ai Agent | NA | 147 (49.00%) | - |
| Gru | NA | 146 (48.67%) | - |
| Globant Code Fixer Agent | NA | 145 (48.33%) | <$1.00 |
| devlo | NA | 142 (47.33%) | - |
| PatchPilot | Claude-3.5-Sonnet | 136 (45.33%) | $0.97 |
| Kodu-v1 | Claude-3.5-Sonnet | 134 (44.67%) | - |
| OpenHands+CodeAct v2.1 | Claude-3.5-Sonnet | 126 (41.67%) | $1.33 |
| Composio SWE-Kit | NA | 124 (41.00%) | - |
| Agentless | Claude-3.5-Sonnet | 123 (40.67%) | - |
| | GPT-4o | 96(32.00 %) | $0.70 |
| Bytedance MarsCode | NA | 118 (39.33%) | - |
| Moatless | Claude-3.5-Sonnet | 115 (38.33%) | $0.30 |
| | GPT-4o | 74(24.67 %) | $0.13 |
| Honeycomb | NA | 115 (38.33%) | - |
| AppMap Navie v2 | NA | 108 (36.00%) | - |
| Isoform | NA | 105 (35.00%) | - |
| SuperCoder2.0 | NA | 102 (34.00% | - |
| Alibaba Lingma Agent | Claude-3.5-Sonnet + Claude-3.5-Sonnet | 99 (33.00%) | - |
| CodeShellTester | GPT-4o | 94(31.33 %) | - |
| Amazon Q Developer-v2 | NA | 89 (29.67%) | - |
| SpecRover | GPT-4o + Claude-3.5-Sonnet | 93 (31.00%) | $0.65 |
| CodeR | GPT-4 | 85 (28.33%) | $3.34 |
| SIMA | GPT-4o | 83 (27.67%) | $0.82 |
| MASAI | NA | 82 (27.33 %) | - |
| IBM Research Agent-101 | NA | 80(26.67 %) | - |
| Aider | GPT-4o + Claude-3.5-Sonnet | 79(26.33 %) | - |
| IBM AI Agent SWE-1.0 | NA | 71(23.67 %) | - |
| Amazon Q Developer | NA | 61 (20.33%) | - |
| AutoCodeRover-v2 | GPT-4o | 92 (30.67%) | - |
| RepoGraph | GPT-4o | 89 (29.67%) | - |
| Openhands+CodeAct v1.8 | Claude-3.5-Sonnet | 80(26.67 %) | $1.14 |
| SWE-agent | Claude-3.5-Sonnet | 69(23.00 %) | $1.62 |
| | GPT-4o | 55 (18.33%) | $2.53 |
| | GPT-4 | 54 (18.00%) | $2.51 |

section and union of resolved instances across the three runs for both tools. PatchPilot achieved a total intersection of 25 instances and a union of 38 instances, yielding a higher intersection-to-union ratio of 0.66. In contrast, OpenHands had a total intersection of 12 instances and a union of 25 instances, with a lower intersection-to-union ratio of 0.48. The larger and more stable intersection instances for PatchPilot indicate its higher consistency in resolving instances across different runs. This stability is further emphasized by the Venn diagram, which visually demonstrates PatchPilot's ability to consistently resolve overlapping instances, underscoring its reliability and effectiveness.

## C.3. Failed Attempts with DeepSeek-r1.

During our experiments with four platforms offering DeepSeek-r1 model services (*DeepSeek* (DeepSeek, 2025), *together.ai* (together.ai, 2024), *firework.ai* (fireworks.ai, 2024), and *deepinfra* (deepinfra, 2024)), we encountered multiple barriers that prevented us from completing the necessary tests. Specifically, DeepSeek's own platform was affected by a DDoS attack, causing repeated service disruptions. Meanwhile, both together.ai and firework.ai exhibited severe response delays that rendered our experiments infeasible. In addition, *deepinfra* restricted the permissible context length for DeepSeek-r1 to 16k tokens, falling short of our PatchPilot pipeline's requirements. Consequently, despite our attempts, we

were unable to obtain experimental results for DeepSeek-r1 under these conditions.

## C.4. Case Study

*Listing 1.* Diff of the issue-introducing commit of django-12284.

```
 1 ||          if not getattr(cls, self.attname, None):
 2 ||              setattr(cls, self.attname, self.descriptor_class(self))
 3 ||      if self.choices is not None:
 4 |-          setattr(cls, 'get_%s_display' % self.name,
 5 |-                  partialmethod(cls._get_FIELD_display, field=self))
 6 |+          if not hasattr(cls, 'get_%s_display' % self.name):
 7 |+              setattr(
 8 |+                  cls,
 9 |+                  'get_%s_display' % self.name,
10 |+                  partialmethod(cls._get_FIELD_display, field=self),
11 ||+              )
```

*Listing 2.* Our best patch in the first batch.

```
 1 ||          if not getattr(cls, self.attname, None):
 2 ||              setattr(cls, self.attname, self.descriptor_class(self))
 3 ||      if self.choices is not None:
 4 |-          if not hasattr(cls, 'get_%s_display' % self.name):
 5 |-              setattr(
 6 |-                  cls,
 7 |-                  'get_%s_display' % self.name,
 8 |-                  partialmethod(cls._get_FIELD_display, field=self),
 9 |-              )
10 |+          setattr(
11 |+              cls,
12 |+              'get_%s_display' % self.name,
13 |+              partialmethod(cls._get_FIELD_display, field=self),
14 ||+          )
```

*Listing 3.* Our final patch for django-12284.

```
 1 ||          if not getattr(cls, self.attname, None):
 2 ||              setattr(cls, self.attname, self.descriptor_class(self))
 3 ||      if self.choices is not None:
 4 |-          if not hasattr(cls, 'get_%s_display' % self.name):
 5 |+          method_name = 'get_%s_display' % self.name
 6 |+          if method_name not in cls.__dict__:
 7 ||              setattr(
 8 ||                  cls,
 9 |-                  'get_%s_display' % self.name,
10 |+                  method_name,
11 ||                  partialmethod(cls._get_FIELD_display, field=self),
12 ||              )
```

*Listing 4.* Golden patch for django-12284.

```
 1 ||          if not getattr(cls, self.attname, None):
 2 ||              setattr(cls, self.attname, self.descriptor_class(self))
 3 ||      if self.choices is not None:
 4 |-          if not hasattr(cls, 'get_%s_display' % self.name):
 5 |+          if 'get_%s_display' % self.name not in cls.__dict__:
 6 ||              setattr(
 7 ||                  cls,
 8 ||                  'get_%s_display' % self.name,
 9 ||                  partialmethod(cls._get_FIELD_display, field=self),
10 ||              )
```

**Successful Cases.** Listing 1– 4 shows an example issue that was resolved by PatchPilot but rarely resolved by other methods in Table 3. Among the top 10 methods, only Blackbox AI Agent successfully solved this issue besides our approach.

Issue details: In Django, when a model field defines *choices*, the *get_xxx_display* method returns the label for a field's value. However, if a subclass overrides the *choices*, the *get_xxx_display* method may not work correctly for the new choices. This happens because Django reuses the method from the parent class instead of creating a new one for the subclass, leading to incorrect results for the updated choices. The patch solution should ensure that Django generates a new *get_xxx_display* method specifically for the subclass to properly handle the overridden choices.

Findings: In our study of processing this issue, we discovered that PatchPilot 's two core features are critical to resolving the issue: *(1) pinpointing the issue-introducing commit* to accurately localize the root cause and gain insight into the critical logic modifications directly related to the issue. and *(2) leveraging feedback from failed tests* to iteratively refine the patch until all tests are passed.

Listing 1 shows the diff of the issue-introducing commit, which closely matches the code area modified by the golden patch in Listing 4 and clearly shows the logical changes that led to the issue.

In the first batch of patching, the best patch generated by PatchPilot is shown in Listing 2. It patched the buggy code by directly reversely applying the diff in the Listing 1. This patch successfully fixes the buggy behavior; however, original functionalities are also affected, leading to a failed functionality test.

In the second batch, PatchPilot retrieves the output of the failed functionality test and its code, performs refinement based on the patch in Listing 2. PatchPilot successfully fixes the broken functionality and generates a qualified patch that passes all tests, as illustrated in Listing 3, which is semantically equivalent to the golden patch shown in Listing 4.

This case study demonstrates the effectiveness of our proposed techniques in Section 3.

**Failed Cases.** Listing 5 shows an vulnerable code chunk in the widely used `scikit-learn` package (Pedregosa et al., 2011). Lines 3 and 6 raise an IndexError because `best_indices[i]` may exceed the size of the second or third dimension of `coefs_paths`. After feeding

*Listing 5.* Example of a vulnerable code chuck.

```
1  best_indices = np.argmax(scores, axis=1)
2  if self.multi_class == 'ovr':
3      w = np.mean([coefs_paths[i, best_indices[i], :]
4                  for i in range(len(folds))], axis=0)
5  else:
6      w = np.mean([coefs_paths[:, i, best_indices[i], :]
7                  for i in range(len(folds))], axis=0)
```

this chuck with exact vulnerable line numbers and the correct bug description into the SOTA GPT-4o and Claude-3.5-Sonnet models, both suggested a plan to constrain the indexing on Lines 3 and 6 using the modulo operations to prevent IndexError. Following this plan, the models generated patches that change `best_indices[i]` in Line 3 and 6 to `best_indices[i]%coefs_paths.shape[1]` and `best_indices[i]%coefs_paths.shape[2]`. Although this patch avoids the IndexError, it also breaks the original computation logic of the program. This example demonstrates that SOTA LLMs still fall short in understanding program logic and reasoning about vulnerabilities. Without a deep understanding, these models tend to propose simple patches that either cannot resolve the problem or harm the functionalities.

# D. Prompts

## D.1. Standard-Patch Prompt

**System Prompt**:
You are an experienced software maintainer responsible for analyzing and fixing repository issues. Your role is to:
1. Thoroughly analyze bugs to identify underlying root causes beyond surface-level symptoms.
2. Provide clear, actionable repair plans with precise code modifications.

Format your repair plans using:
- <STEP> and </STEP> tags for each modification step.
- <Actions to be Taken> and </Actions to be Taken> tags for specific actions.
- Maximum 3 steps, with each step containing exactly one code modification.
- Only include steps that require code changes.

If the issue text includes a recommended fix, do not apply it directly. You should explicitly reason whether it can fix the issue. Output the reason that the recommended fix can or cannot fix the issue. You should explicitly reason whether the recommended fix keeps the same code style and adapt it to align with the codebase's style and standards. Ensure that the patch considers interactions across different code sections, including nested structures, function calls, and data dependencies.
The patch should maintain overall structural integrity, addressing the issue without unintended effects on other parts. Propose solutions that are resilient to structural changes or future extensions.

**User Prompt**:
You are required to propose a plan to fix a issue.
Follow these guidelines:
- Number of Steps: The number of steps to fix the issue should be at most 3.
- Modification: Each step should perform exactly one modification at exactly one location in the code.
- Necessity: Do not modify the code unless it is necessary to fix the issue.
Your plan should outline only the steps that involve code modifications. If a step does not require a code change, do not include it in the plan.
Do not write any code in the plan.

`{format_example}`

Here is the issue text:
— BEGIN ISSUE —
`{problem_statement}`
— END ISSUE —

Below are some code segments, each from a relevant file. One or more of these files may contain bugs.
— BEGIN FILE —
`{content}`
— END FILE —

`{feedback_from_failed_tests}`

## D.2. Minimal-Patch Prompt

The following prompt was used to generate a plan that fixes the issue with minimal modification. The additional prompt, compared to the standard-patch prompt in Appendix D.1, is highlighted in blue.

**System Prompt**:
You are an experienced software maintainer responsible for analyzing and fixing repository issues. Your role is to:
1. Thoroughly analyze bugs to identify underlying root causes beyond surface-level symptoms.
2. Provide clear, actionable repair plans with precise code modifications.
3. In fixing the bug, your focus should be on making minimal modifications that only target the bug-triggering scenario without affecting other parts of the code or functionality.
4. Make sure that other inputs or conditions are not impacted. Modify only the specific behavior causing the bug, and do not make any broad changes unless absolutely necessary.
Format your repair plans using:
- <STEP> and </STEP> tags for each modification step.
- <Actions to be Taken> and </Actions to be Taken> tags for specific actions.
- Maximum 3 steps, with each step containing exactly one code modification.
- Only include steps that require code changes.

If the issue text includes a recommended fix, do not apply it directly. You should explicitly reason whether it can fix the issue. Output the reason that the recommended fix can or cannot fix the issue. You should explicitly reason whether the recommended fix keeps the same code style and adapt it to align with the codebase's style and standards. Ensure that the patch considers interactions across different code sections, including nested structures, function calls, and data dependencies.
The patch should maintain overall structural integrity, addressing the issue without unintended effects on other parts. Propose solutions that are resilient to structural changes or future extensions.

**User Prompt**:
You are required to propose a plan to fix a issue.
Follow these guidelines:
- Number of Steps: The number of steps to fix the issue should be at most 3.
- Modification: Each step should perform exactly one modification at exactly one location in the code.
- Necessity: Do not modify the code unless it is necessary to fix the issue.
- One File Only: Choose one file to modify, and ensure all changes are limited to that file.
- Concentration on Input: If the issue mentions a specific input or argument that triggers the bug, ensure your solution only fixes the behavior for that input.
Your plan should outline only the steps that involve code modifications. If a step does not require a code change, do not include it in the plan.
Do not write any code in the plan.

`{format_example}`

Here is the issue text:
— BEGIN ISSUE —
`{problem_statement}`
— END ISSUE —

Below are some code segments, each from a relevant file. One or more of these files may contain bugs.
— BEGIN FILE —
`{content}`
— END FILE —

## D.3. Comprehensive-Patch Prompt

The following prompt was used to generate a plan that comprehensively fixes the issue. The additional prompt, compared to the standard-patch prompt in Appendix D.1, is highlighted in red.

**System Prompt**:
You are an experienced software maintainer responsible for analyzing and fixing repository issues. Your role is to:
1. Thoroughly analyze bugs to identify underlying root causes beyond surface-level symptoms.
2. Provide clear, actionable repair plans with precise code modifications.
3. The plan should ensure that the solution is general, preventing any similar bugs from occurring in other contexts or input variations.
4. The plan will be used to generate a comprehensive and extensive patch that addresses the issue thoroughly, modifies related areas to ensure the bug is fully resolved, and enhances the overall robustness and completeness of the solution.
Format your repair plans using:
- <STEP> and </STEP> tags for each modification step.
- <Actions to be Taken> and </Actions to be Taken> tags for specific actions.
- Maximum 3 steps, with each step containing exactly one code modification.
- Only include steps that require code changes.

If the issue text includes a recommended fix, do not apply it directly. You should explicitly reason whether it can fix the issue. Output the reason that the recommended fix can or cannot fix the issue. You should explicitly reason whether the recommended fix keeps the same code style and adapt it to align with the codebase's style and standards. Ensure that the patch considers interactions across different code sections, including nested structures, function calls, and data dependencies.
The patch should maintain overall structural integrity, addressing the issue without unintended effects on other parts.
Propose solutions that are resilient to structural changes or future extensions.

**User Prompt**:
You are required to propose a plan to fix a issue.
Follow these guidelines:
- Number of Steps: The number of steps to fix the issue should be at most 3.
- Modification: Each step should perform exactly one modification at exactly one location in the code.
- Necessity: Do not modify the code unless it is necessary to fix the issue.
- Broad Solution: Your solution should aim to resolve the issue broadly and comprehensively, covering edge cases and general patterns.
- No assumptions about input: Avoid making assumptions based solely on the example given in the issue description. If the issue description mentions specific cases (e.g., a particular input or argument), the fix should be applicable to all possible cases, not just the ones listed.

{format_example}

Here is the issue text:
— BEGIN ISSUE —
{problem_statement}
— END ISSUE —

Below are some code segments, each from a relevant file. One or more of these files may contain bugs.
— BEGIN FILE —
{content}
— END FILE —

