# OpenReview forum: "PatchPilot: A Cost-Efficient Software Engineering Agent with Early Attempts on Formal Verification"
_ICML.cc/2025/Conference — ICML 2025 poster_

### Official Review · Reviewer_Sjbd · 2025-03-05

**Overall Recommendation:** 4

**Summary:**

This paper presents an improvement to the "Agentless" approach to solving SWE-bench tasks. They manager to solve 3-5% more problems on SWE-bench Lite/Verified while using up to 20% *less* money, with Claude 3.5. They provide detailed analysis and ablations.

**Claims And Evidence:**

Yes, there is a wide array of experiments that were run in this work to support the claims.

**Essential References Not Discussed:**

None.

**Experimental Designs Or Analyses:**

Yes, as detailed in the SWE-bench guide, the only major issues that can happen with SWE-bench evaluations is either submissions using the 'hints' column which should not be used, or submissions using knowledge about fail2pass or pass2pass tests before submitting. I saw no evidence for either of these things being done in this work.

**Methods And Evaluation Criteria:**

Yes SWE-bench is the leading method to test software engineering agents, and obtaining *higher* SWE-bench scores at *lower* cost is a sign of a substantial improvement.

**Other Comments Or Suggestions:**

I would maybe not call it 'human-based planning'. Kinda makes it seem like you have a human-in-the-loop "Devin" like system where the human is planning and the agent is executing. Can you find a phrase for this that doesn't have the word 'human' maybe?

**Other Strengths And Weaknesses:**

I think this paper is very strong overall: great results, easy to read, good analysis, very interesting discussion section.

**Questions For Authors:**

Is your system's wall-time better than Agentless?

**Relation To Broader Scientific Literature:**

There's two lines of work on how to solve swe-bench-style tasks: SWE-agent-like models and Agentless-like models. The first usually get higher scores but are very expensive, the Agentless models usually do slightly worse but are very cheap. This paper is a clear contribution to the second type of models, which are very important.

**Theoretical Claims:**

n/a

---

> ### Author Rebuttal · Authors · 2025-03-31
>
> Thanks for the constructive and positive comments.
>
> ## 1. Extra Clarification on Terminology: "Human-Based Planning"
>
> We thank the reviewer for pointing this out. We agree that the term "human-based planning" may be misleading, as it could imply a human-in-the-loop system. To avoid this confusion, we will revise it to "Rule-based planning", which more accurately reflects our intent—i.e., that the plan is rules pre-specified by developers.
>
> ## 2. Wall-Time Comparison with Agentless
>
> We followed the reviewer’s suggestions and conducted an experiment to compare the wall-time of our system with Agentless and OpenHands on 100 cases (the selection of these cases is stated in the response-1 to reviewer Lckx). Since all tools support multiprocessing, we controlled the number of processes to be the same for all methods (8) to ensure a fair comparison. We measured the average time per instance and ran the experiment three times. The results are as follows.
>
> | Stability test | Round 1 |  | Round 2 |  | Round 3 |  | Average Time |
> |----------------|---------|--------|---------|--------|---------|--------|--------------|
> | PatchPilot | 38/100 | 85min | 38/100 | 86min | 39/100 | 89min | 87 min |
> | Openhands | 31/100 | 86min | 35/100 | 85min | 39/100 | 99min | 90 min |
> | Agentless | 35/100 | 102min | 33/100 | 103min | 34/100 | 100min | 102 min |
>
> The results show that PatchPilot is faster than agentless and OpenHands. It also shows that PatchPilot is more effective than the baseline approaches.

---

### Official Review · Reviewer_vPVX · 2025-03-14

**Overall Recommendation:** 2

**Summary:**

The paper proposes PatchPilot, an agentic patching framework designed to address the trade-offs among patching efficacy, stability. It introduces a novel human-based planning workflow, incorporating 6 components, with special emphasis on refinement as a unique contribution.

**Claims And Evidence:**

Their claims, supported by extensive empirical evidence and rigorous evaluation on SWE-Bench performance.

**Essential References Not Discussed:**

Should add more related work.

**Experimental Designs Or Analyses:**

Authors could enhance the analysis by including more intuitive visualizations, such as detailed plots illustrating trade-offs in stability and cost. Why it surpassed agentless in a clear way.

**Methods And Evaluation Criteria:**

Yes.

**Other Comments Or Suggestions:**

Could you elaborate on how PatchPilot specifically addresses or mitigates the potential instability caused by iterative refinement?

The manuscript could provide a deeper discussion of the scenarios where PatchPilot may fail or perform sub-optimally.

**Other Strengths And Weaknesses:**

Strengths: See above

Weaknesses:

The performance is good, but the authors should provide more evidence to support their claims, as their method uses fewer tokens yet achieves good results. However, the experimental analysis does not fully substantiate their claims and lacks several important insights.

**Questions For Authors:**

See above

**Relation To Broader Scientific Literature:**

N/A

**Theoretical Claims:**

Not much theoretical claims.

---

> ### Author Rebuttal · Authors · 2025-03-31
>
> Thanks for the constructive and positive comments.
>
> ## 1. Better experiment analysis
>
> We will follow the reviewer’s suggestion and include more visualizations to show the comparison between our method and baselines in effectiveness, stability, and cost. To better demonstrate the advantage of PatchPilot over Agentless, we reran these two methods on 100 cases three times (the selection of these cases is stated in response-1 to reviewer Lckx) and reported their resolved rate, average cost, and run time.
> |   | Round 1          | Round 2          | Round 3          | Average resolved | Average cost         | Average time |
> |--------|------------------|------------------|------------------|--------------|-------------|-----------|
> | PatchPilot | 38/100 ($98.23)  | 38/100 ($99.60)  | 39/100 ($97.88)  |    38.33   | 98.57 | 87 min   |
> | Agentless  | 35/100 ($117.75) | 33/100 ($120.31) | 34/100 ($117.02) |   34   | 118.36 | 102 min |
>
> The results show that PatchPilot consistently outperforms Agentless in all three metrics.
>
>
> ## 2. Mitigating Potential Instability Introduced by Iterative Refinement
>
> Thanks for pointing this out. We incorporate two strategies to improve stability during refinement.
>
> **(1) Batch-based Refinement with Selection Mechanism:** Instead of generating a single patch per iteration by directly modifying the previous one, PatchPilot generates multiple patch candidates in each batch based on the previous best patch. It then evaluates all candidates—together with the previous best patch—using both the PoC and functionality tests and selects the best-performing one. If all current candidates underperform compared to the previous best patch, the previous best patch is resolved. This mechanism significantly reduces the risk of accumulated generation noise and protects against overwriting correct earlier solutions (validated by our hyper-parameter experiment when we reduce the batch size from 4 to 1, the overall performance drops notably).
>
> **(2) Early Stopping:** As soon as a patch passes all tests, iterative refinement is terminated immediately, and the patch is returned. This prevents unnecessary iterations without useful feedback, avoiding potential degradation from further blind refinement.
>
> ## 3. Failure Cases of PatchPilot
>
> We will include more analysis of failure cases. In summary, some of the failure cases are because the issue descriptions are of low quality, such as incomplete or incorrect descriptions of the issues. To resolve these cases, we need to rewrite high-quality issue descriptions. Some other cases are shallow patches, where PatchPilot only patches the superficial logic instead of addressing the root cause and does not consider the corner case of the issue, or over-patching, where patches inadvertently affect more scenarios than necessary, causing unexpected behavior. These cases may be further resolved by fine-tuning specific models that can better understand the program logic of the target repos.

---

### Official Review · Reviewer_HrpY · 2025-03-19

**Overall Recommendation:** 2

**Summary:**

This paper proposes PatchPilot, an agentic framework for autonomous software patching. It relies on human-based planning and consists of five workflow components: reproduction, localization, generation, validation, and refinement. The overall workflow as well as each component (except for the final refinement step) closely resembles prior work Agentless. The key difference being:

- During generation, PatchPilot explicitly prompts the model to output a multi-step patching plan.
- Instead of regenerating patches, PatchPilot refines existing ones based on validation feedback.
- Prompt engineering: The model is guided to generate diverse patches by explicitly prompting for both simple and complex solutions.

Experiments conducted on SWE-bench (Lite and Verified) show that PatchPilot outperforms SOTA open-source methods while having the lowest cost. Results also show that PatchPilot is more stable than the SOTA agent-based planning method OpenHands.

**Claims And Evidence:**

The paper presents an incremental improvement rather than a novel contribution, integrating existing tools and iterative refinement workflows from prior agentic work into the Agentless workflow without adequately acknowledging previous research.

**Overall Design**

The approach closely follows the Agentless workflow, with the first four stages - reproduction, localization, generation, and validation - remaining largely unchanged. The primary modifications, such as searching tools, are also borrowed from prior work and are not novel.

The paper claims the Refinement is a unique component to PatchPilot, but this is not the case. Early LLM-based automated program repair work, such as ChatRepair [a], has already demonstrated that test execution feedback can refine previous patches more efficiently than regenerating them from scratch. Multi-agent frameworks like MarsCode [b] and SpecRover [c] similarly refine the generated patch based on test execution results. Also, existing agentic tools like OpenHands [d] enable LLMs to autonomously refine the PoC reproduction scripts and patch during their interactions - an approach commonly seen in their repair trajectories.


**Other Technical Designs Inspired by Prior Work**

The other technical designs also closely resemble prior work from the same research area; however, the authors do not acknowledge these contributions and instead present them as if they were their own novel designs.

- Search tools: Similar tools have been proposed in AutoCodeRover [e] and CodeR [f], yet the authors did not acknowledge these prior contributions and present them as if they were their own novel design.

- Coverage information: AutoCodeRover [e] optionally leverages coverage analysis, such as Spectrum-based Fault Localization, for context retrieval.

- Separating planning and generation: The separation of planning has been explored in prior work, such as AppMap Navie [g].

- Patch validation strategy: The strategy is nearly identical to Agentless, except for prioritizing patches that pass PoC tests over functionality tests, whereas Agentless prioritizes patches that pass functionality tests over PoC tests. However, the authors do not clarify that this strategy is a modification of existing work.


[a] Keep the Conversation Going: Fixing 162 out of 337 bugs for $0.42 each using ChatGPT. (ISSTA 2024)

[b] MarsCode Agent: AI-native Automated Bug Fixing. (ArXiv 2024)

[c] SpecRover: Code Intent Extraction via LLMs. (ICSE 2025)

[d] OpenHands: An Open Platform for AI Software Developers as Generalist Agents. (ArXiv 2024)

[e] AutoCodeRover: Autonomous Program Improvement. (ISSTA 2024)

[f] CodeR: Issue resolving with multi-agent and task graphs. (ArXiv 2024)

[g] AppMap Navie: https://appmap.io/blog/2024/06/20/appmap-navie-swe-bench-leader/

**Essential References Not Discussed:**

Essential related works are cited but not adequately discussed in the paper. (see above)

**Experimental Designs Or Analyses:**

In Section 4.2, the authors compare PatchPilot with OpenHands on a subset of 45 problems, repeating the experiments three times and reporting standard deviation. However, I wonder if this is statistically significant enough to support a strong conclusion.

**Methods And Evaluation Criteria:**

There are many hyperparameters that have not been examined through ablation studies, such as the iteration limit for refinement, the number of retrieved files, the number of plans and patches for a single instance. It is also unclear how sensitive the technique's performance and cost are to these parameters.

Moreover, some key hyperparameters, such as the sampling temperature, are not even clearly specified.

Update after rebuttal:

* Ablation study shows that prompt engineering like the diverse-prompting design may be a main driver of improvement. I am not fully convinced by the main claim of this paper.

**Other Comments Or Suggestions:**

- “SWE-Bench” should be “SWE-bench”

**Other Strengths And Weaknesses:**

Other strengths:

- Outperforms SOTA open-source methods while having the lowest cost

**Questions For Authors:**

1. How much improvement does prompt engineering contribute (specifically, by using three different prompts to generate simple, complex, and standard fixes) ? The ablation study did not isolate this design.

**Relation To Broader Scientific Literature:**

Overall, the contribution is insignificant compared to prior literature:

- The proposed technique mainly integrates agentic designs (such as tool use and iterative refine loops) into an agentless framework, offering little novelty.

- Observations on the cost-effectiveness and stability of human-based planning versus agent-based planning have been made before, as have insights on the effectiveness of refining existing patches compared to generating new ones.

**Theoretical Claims:**

N/A

---

> ### Author Rebuttal · Authors · 2025-03-31
>
> Thanks for the constructive comments.
>
> ## 1. Novelty and Differences from Existing Tools
>
> As discussed in Section 2, [a-g] are all agent-based, differing from our workflow (we included [b-g] and will add [a]).
> - Search tools: We acknowledge that AutoCodeRover and CodeR also have search tools, and we are inspired by their designs. The key differences are in the search mechanisms, where we use fuzzy search instead of exact search. As the searched strings may differ from the original code segments, the exact search misses some matches.
> - Coverage: AutoCodeRover's SBFL method requires post-patching test cases, which are unavailable in SWE-bench problem settings. PatchPilot generates coverage using only PoCs.
> - Separating planning and generation: We respectfully point out that AppMap Navie is a commercial product without research papers. Although it mentions having generation plans, the technical details are not available. Our separation design was inspired by our early observations of shallow patches. Besides, the detailed design can be very different. For example, we proposed diversity prompts and step-by-step generation. We believe it is reasonable to claim the novelty and originality of our design of the generation component.
> - Patch validation vs. Agentless: Agentless generates assertions for testing cases and validates based on assertions. It is hard to generate correct and comprehensive assertions. PatchPilot generates only PoCs and uses LLMs to validate based on PoC outputs. We applied both methods to 100 patches (50 from PatchPilot, 50 from Agentless). Our method identified 43 correct patches; Agentless identified 40.
> - PatchPilot differs from Agentless in the workflow (**refinement**) and applies optimizations to each component. Although the components' high-level ideas are similar, the specific designs can vary a lot and affect final performance, as shown in the comparison with Agentless.
> - About refinement: Agent-based tools refine the latest patch by prompting the LLM with prior conversation history. This simple method has a few limitations and cannot be applied to PatchPilot: 1) It generates only one patch in the first round and iteratively refines it, which lacks diversity. If the initial patch contains errors, it may be hard to fix. This is partially validated in our hyper-parameter experiments with a batch size of 1, showing a performance drop. 2) It cannot revert to earlier patches if a refinement worsens the result, while PatchPilot enables such a capability. 3) By including the refinement history in prompts, it risks context overflow and potential LLM confusion from prior bad patches. PatchPilot instead uses only the best previous patch and its validation results. Given these major differences, we believe it is reasonable to claim novelty for our refinement.
>
> While existing tools may share similar high-level ideas, our design is fundamentally different and more advanced, giving better performance and stability. We believe this brings essential novelty and contribution.
>
> ## 2. Extra Ablation Studies for Hyperparameters
>
> We conducted experiments to evaluate hyper-parameter sensitivity. PatchPilot includes the following hyper-parameters (default values in brackets): number of retrieved files (5), total patches generated (12), batch size per round (4), diversity prompts (3), and generation temperature (0.8). Each experiment varies one hyper-parameter while keeping others. We used 60 cases randomly selected from SWE-bench-Lite and Claude-3.5-Sonnet.
> - Num. of retrieved files in localization: We changed it from 5 to 3 and 7. Varying it introduces subtle differences, but retrieving more files will have a higher cost as LLM will have more conversations and longer contexts.
> - Total num. of patches: Changing it from 12 to 1, 4, 8, 16. A large one slightly improves performance but raises costs. 12 strikes a balance between performance and cost.
> - Batch Size: Changing the batch size from 4 to 2/6 introduces performance drops. For 2, the diversity is not enough; for 6, the refinement round (2) is not enough. A moderate batch size balances the refinement rounds and diversity.
> - Diversity Prompts: Instead of using three prompts together, we ran the experiment with each prompt separately. Mixing them gives the best performance as it enables the highest diversity.
> - We varied the temperature ($\tau$) from 0.8 to 1 and 0.6. The results are similar. We set it high to encourage diversity.
>
> | Variants|Number|Resolved|Cost ($)|
> |---------|---------|-----------|----------|
> | Default| - |28/60|58.27|
> |Retrieved files| 3|24/60|50.38|
> |   | 7|27/60|64.23|
> |#. patches|1|20/60|30.96|
> | | 4|26/60|39.95|
> | |8|27/60|51.02|
> | |16|28/60|62.88|
> | Batch| 2|24/60|62.93|
> | |6|25/60|55.77|
> |Prompt|Standard|26/60|57.67|
> | | Big| 25/60| 61.17|
> | | Small| 22/60| 55.32|
> |$\tau$| 0.6|27/60| 58.21|
> | | 1.0|28/60|58.02|
>
> ## 3. Stability Evaluation
>
> Our stability experiment is in **response-1 to reviewer Lckx.**

---

> > ### Comment · Reviewer_HrpY · 2025-04-07
> >
> > Thanks for the response.
> >
> > I remain unconvinced by the claim of novelty of this paper. For example, I wouldn’t consider fuzzy search to be “fundamentally different and more advanced” from exact search. Please also kindly note that fuzzy match has already been used in existing work like MISAI [a], which has a research paper. Similarly, I would not consider the patch validation or refinement design to be “fundamentally different and more advanced” compared to prior agentic-based work.
> >
> > > AutoCodeRover's SBFL method requires post-patching test cases, which are unavailable in SWE-bench problem settings. PatchPilot generates coverage using only PoCs.
> >
> > Please note that the follow-up work of AutoCodeRover, CodeR [b], overcomes this limitation by using self-generated PoCs to get coverage information (instead of PoC, they call it “reproduced test cases” generated by a Reproducer agent). Despite this slight modification, the method is still acknowledged as SBFL. PatchPilot’s strategy is just a simplified version.
> >
> > > Separating planning and generation: We respectfully point out that AppMap Navie is a commercial product without research papers. Although it mentions having generation plans, the technical details are not available.
> >
> >
> > Please kindly note that AppMap Navie was open-sourced in June 2024 [c][d], and its technical blog [e] has been cited and discussed by papers such as SpecRover [f] and Agentless. From the SpecRover paper: “Navie uses a retrieval-augmented generation (RAG) based approach to construct the code context, and performs an explicit planning step before generating code changes.”
> >
> > If the step-by-step planning is a major novel contribution of this paper, an ablation study should be provided to support the claim, and the authors should, at the very least, cite this line of work (AppMap Navie, CodePlan [g], etc) and discuss the differences.
> >
> >
> > > diversity prompts
> >
> >
> > The simple idea of using diverse prompts - i.e., three different hand-crafted prompts to generate standard, minimal, and maximal edits - is indeed a novel and effective design of PatchPilot, as evidenced by the ablation study. I already acknowledged this in my original review. Prior work such as MASAI [a] uses similar prompt engineering (i.e., prompting the LLM for minimal rewrites), but only applies a single style of prompting, not multiple.
> >
> > However, the proposed prompt engineering is somewhat ad hoc. For instance, PatchPilot’s minimal-patch prompt includes:
> >
> > * A restriction to “modify one file only”, while some issues require multi-file edits. The exact prompt reads:
> >
> > > “One File Only: Choose one file to modify, and ensure all changes are limited to that file”.
> >
> > * Fix only the specific input mentioned in the issue, potentially preventing general fixes. The exact prompt reads:
> >
> > > “If the issue mentions a specific input or argument that triggers the bug, ensure your solution only fixes the behavior for that input”.
> >
> > While the direction of using prompt engineering to increase patch diversity is interesting and promising, I do not find the current approach to be systematic or novel enough to justify acceptance.
> >
> > Also, according to the new ablation results, this three-prompt design yields an approximately improvement of (28-26)/26=7.7% relative to standard prompting. The default PatchPilot solves 136 problems, outperforming Agentless (which uses standard prompting and solves 123 problems) with a relative improvement of (136-123)/123=10.6%. This might suggest that:
> >
> > * Without this prompt engineering, PatchPilot could solve ~126 problems (136 / 1.077), v.s. Agentless’s 123.
> >
> > * Conversely, Agentless with prompt diversity might be capable of solving ~132.5 problems (123 * 1.077).
> >
> > Disclaimer: I understand that the ablation result may be imprecise due to the small sample size. I just use the 7.7% number here for illustrative purposes.
> >
> > In other words, although this paper claims a number of novel contributions - including refinement - I am not fully convinced, as prompt engineering like this three-prompt design may be the main driver of improvement rather than any fundamentally new methodology.
> >
> > In summary, the authors’ response does not sufficiently address my concern. As reviewer Lckx also notes, the novelty is low. I continue to lean towards rejection.
> >
> > References:
> >
> > [a] MASAI: Modular Architecture for Software-engineering AI Agents
> > [b] CodeR: Issue resolving with multi-agent and task graphs. (ArXiv 2024)
> >
> > [c] https://github.com/SWE-bench/experiments/issues/28
> >
> > [d] https://github.com/getappmap/SWE-bench
> >
> > [e] AppMap Navie: https://appmap.io/blog/2024/06/20/appmap-navie-swe-bench-leader/
> >
> > [f] SpecRover: Code Intent Extraction via LLMs. (ICSE 2025)
> >
> > [g] CodePlan: Repository-level Coding using LLMs and Planning. (FSE 2024)

---

> > > ### Author Response · Authors · 2025-04-07
> > >
> > > Thank the reviewer for the additional questions.
> > >
> > > First, we respectfully point out that PatchPilot achieved the *highest resolved rate* on the SWE-bench-Lite and SWE-bench-Verified benchmarks among all open-sourced tools when we submitted it. It is *most stable and cost-efficient* among the top-performing tools. Given the internal statistical nature of LLMs and the high cost of their APIs, achieving high resolved rate, stability, and low cost together is notably challenging and requires non-trivial efforts.
> > >
> > > Second, we emphasize that although some existing works have a similar high-level idea to us, the technical designs are different. For example, existing agent-based planning work has refinement, which is not directly applicable to our system given the different workflow. Besides, the actual refinement mechanisms are different, as our refinement needs to be highly coupled and integrated with our other components. We believe this does not dilute our technical contributions, given the difference of actual designs and improvements of our tool in performance, stability, and cost-efficiency. Given their relatively low performance, simply following their designs would not lead to a performance improvement.
> > >
> > > Third, we respectfully point out that by saying “our design is fundamentally different and more advanced” we refer to that adding all individual designs together makes our method fundamentally different, and our performance makes it more advanced. We do not claim that *every individual design* is fundamentally different. For example, we did not claim fuzzy search as fundamentally different in the rebuttal. Sorry for the confusion on this.
> > >
> > > Fourth, about fuzzy search, MASAI’s fuzzy search is used in patch generation to find the original code that needs to be replaced by the patch, while we used it in localization, which are not the same. About coverage information, thanks for pointing CodeR out. Our coverage filter is simpler, but it is faster and gives reasonable localization performance.
> > >
> > > About generation, AppMap Navie is different from us. 1) Input to planning: Beyond localization results and issue descriptions (which was used in AppMap), PatchPilot also leverages runtime feedback from both PoCs and functionality tests. 2) Prompting strategy: As acknowledged by the reviewer, PatchPilot employs diverse prompting strategies to generate multiple plan variants. 3) Plan-to-patch execution: PatchPilot performs step-by-step patch generation, where each plan step corresponds to one edition with syntax and linting checks running before generating the next. AppMap Navie directly feeds the plan as part of the prompts for generation. To quantify the impact of these differences, we constructed a variant in which all three design choices were replaced with those of AppMap Navie. On a subset of 150 cases from SWE-bench-Lite, the original PatchPilot vs this variant: 69 vs. 64, showing the importance of our designs. CodePlan targets different problems from us.
> > >
> > > We note that CodeR, MASAI, and AppMap are not accepted papers and are not ranked among the top. We did not include a detailed comparison with their individual components and mainly compared OpenHands. We believe it is a reasonable practice. We apologize for not comparing their design in detail and will add this to our paper.
> > >
> > > About the diversity prompt, we respectfully point out that it is one but not the main contribution. Our ablation study shows that each component contributes to a 3–4% performance gain. We agree that minimal-patch prompts cannot handle all cases. Its goal is to increase diversity rather than handling all cases. We have standard and comprehensive prompts. We respectfully point out that the way to calculate the hypothetical improvement of diversity prompts may not be comprehensive. First, it relies on an assumption that cases contributing to performance improvements by each design are evenly distributed across the dataset, which may not hold as the performance improvements are typically because of joint efforts. Second, our strategy cannot be directly applied to Agentless as it does not generate patch plans. Following the reviewer's idea, we changed “generate plan” to “generate patches” and applied it to Agentless. We ran the original agentless and this variation on 150 cases above. The original Agentless vs this variant: 61 vs. 63. (1) It shows marginal improvements on Agentless. Our design improving an SOTA method does not dilute our contribution. (2) PatchPilot still outperforms this improved Agentless, showing the importance of our other designs. (3) We reran the hyper-parameter testing on 150 cases above, and the resolved cases are 66 (only standard), 64 (only comprehensive), and 59 (only minimal), while combined prompts achieved 69. This shows that the percentage cannot be directly applied to more cases given the cases affecting the results may not be distributed evenly.
> > >
> > > We are at the reviewer’s disposal for any further questions.

---

### Official Review · Reviewer_Lckx · 2025-03-21

**Overall Recommendation:** 3

**Summary:**

In this paper, the authors describe PatchPilot, a novel human-based planning workflow for solving Github issues. The innovations include generating reproduction tests to help locate the root cause; a planning and generation task division for patch generation, and a refinement loop to iteratively improve a patch. Empirical evaluations on SWE-Bench showcases PatchPilot’s efficiency and stability in resolving Github issues compared to baseline methods.

**Claims And Evidence:**

The claims are well-supported by clear evidence. The authors provided overall performance evaluations on SWE-Bench Lite and Verified set. Table 1 shows that PatchPilot strikes a balance between resolution rate and cost. Section 4.3 provided useful information on each component.

**Essential References Not Discussed:**

N/A

**Experimental Designs Or Analyses:**

Most experimental designs are sound. I have one comment for Section 4.2, why did the authors present the PatchPilot vs OpenHands comparison results with GPT-4o instead of Claude 3.5 Sonnet? Considering both methods are reported with Claude 3.5 Sonnet in Table 1, this mismatch should have some explanation.

**Methods And Evaluation Criteria:**

SWE-Bench is a standard benchmark dataset for measuring software agents so it makes sense the authors chose it to validate their work.

**Other Comments Or Suggestions:**

N/A

**Other Strengths And Weaknesses:**

See sections above.

**Questions For Authors:**

1. Can the authors provide an example plan generated for the patch generation step? And a corresponding example with LLM following the generated plan.

2. As the PoC is used to help locate the root cause, its validity is crucial. Besides including instructions in the prompt, did the authors carry out other mechanisms to verify their validity?

3. Majority voting is mentioned in Section 4.3, can the authors provide more information on how the majority voting is carried out?

**Relation To Broader Scientific Literature:**

Key novelties are: using reproduction tests to help localization; planning and generation separation for patch generation, and patch refinement. Some versions of the first two are present in other agent based works such as OpenHands. One could argue OpenHands' iterative approach has the refinement element too as a subset of history is retained in prompts. So the significance of these novelties are not super strong.

**Theoretical Claims:**

N/A

---

> ### Author Rebuttal · Authors · 2025-03-31
>
> Thanks for the positive and constructive comments.
>
> ## 1. Stability Comparison
>
> We chose GPT-4o because of our budget limits before. We reran the stability comparison. First, we changed GPT-4o to Claude-3.5-Sonnet. Second, in response to Reviewer HrpY’s concern about small sample size, we increased the testing cases from 45 to 100 from SWE-bench-Lite. These cases were selected from three categories based on these methods’ results in one previous run: 30 are commonly resolved by all tools, 15 are solved independently by each of the three tools, and 55 are commonly unresolved cases (hard ones). The cases in each category are randomly selected. Third, we added Agentless based on Reviewer vPVX’s suggestion. We calculated the standard deviation (STD) of the resolved rate and cost. To show the statistical significance, we conducted a Bartlett’s test to compare the variance between PatchPilot and the comparison baselines. Our Null hypothesis $H0: \sigma^2_p = \sigma^2_b$,  $\sigma^2_p$ and $\sigma^2_b$ are the variance of PatchPilot and baseline. The results are as follows (the first number in p-value is the comparison of resolved rate, and the other is the cost).
>
> |      | Round 1  | Round 2  | Round 3  | Average Time | STD  | $p$-value |
> |------------|------------------|------------------|------------------|--------------|-------------|-----------|
> | PatchPilot | 38/100 ($98.23)  | 38/100 ($99.60)  | 39/100 ($97.88)  | 87 min  | 0.58($0.91) | -  |
> | OpenHands  | 31/100 ($156.58) | 35/100 ($160.38) | 39/100 ($169.58) | 90 min | 4.00($6.68) | 0.044/0.040 |
> | Agentless  | 35/100 ($117.75) | 33/100 ($120.31) | 34/100 ($117.02) | 102 min  | 1.00($1.73) | 0.50/0.43 |
>
> The result confirms that PatchPilot has a much lower STD than OpenHands in the resolved rate. Besides, the $p$-value is smaller than 0.05, meaning we can reject the null hypothesis. Given that the STD of PatchPilot is smaller than OpenHands, this shows that PatchPilot is more stable than OpenHands with statistical significance. The resolved rates of all methods are lower than those in Table 1 because we include more hard cases in the 100 cases. The cost shows a similar trend. OpenHands is relatively more cost-effective when using Claude-3.5 compared to using GPT-4o in Section 4.2, as a stronger model can reduce the number of iterations of OpenHands.
>
> The result also shows the stability of Agentless, validating our claim that human-based planning is more stable than agent-based planning. Finally, PatchPilot consistently achieves lower cost and higher performance than Agentless.
>
> ## 2. Illustrative Example of Patch Planning and Generation
>
> We will include more examples. Here, we show one example of django__django-11133. Django is a Python web framework. In this issue, the user reported that HttpResponse failed to handle memoryview objects.
>
> **PatchPilot generated a patch plan with one step:**
>
> <STEP> Add an explicit memoryview check in the content setter </STEP>
> <Actions> In the file django/http/response.py, locate the content setter of HttpResponse. Modify the code so that it first checks if the provided value is an instance of memoryview. If it is, immediately return bytes(value). Otherwise, proceed with the existing check for iterables. </Actions>
>
> **Following the plan, PatchPilot generates a correct patch:**
> ```python
>          # - Python attempts str conversion first
>          # - when self._charset != 'utf-8' it re-encodes the content
> -        if isinstance(value, bytes):
> +        if isinstance(value, (bytes, memoryview)):
>              return bytes(value)
> ```
> ## 3.PoC Validation Beyond LLM Checking
>
> We first convert the issue description into a structured format, extracting the expected behavior and wrong behavior. We also include the following rule-based checks.
>
> **Rule 1:** Discard PoCs that cover fewer than 3 files, as LLMs often create fake buggy functions inside the PoC instead of invoking real project code.
>
>
> **Rule 2:** If a specific exception is mentioned in the issue, we check if the same exception appears in stderr during PoC execution, confirming it triggers the described error.
>
> To evaluate PoC quality, we manually reviewed 100 cases. Our validation filtered out 27 PoCs (20 by LLM, 5 by Rule 1, and 2 by Rule 2). Of the remaining 73 PoCs, 92% are correct. These results confirm the effectiveness of our patch validation and the high quality of generated PoCs.
>
> ## 4. Majority Voting Strategy in Ablation Study
>
> In the first two variants, we use majority voting since patch validation is not feasible without PoCs or functionality tests. We generate multiple patches per case and proceed as follows: Normalize each patch by stripping whitespace, line breaks, and comments; count the frequency of each normalized patch; and select the most frequent one. If there is a tie, we prompt an LLM to choose the best patch.
>
> ## 5. Novelty of Refinement in PatchPilot
>
> We discuss the novelty of our refinement in **response-1 to reviewer HrpY.**

---

### Decision · Program_Chairs · 2025-05-01

**Decision:**

Accept (poster)

**Comment:**

**Summary:** The paper proposes PathPilot with a human-based planning workflow composed of five components of reproduction, localization, generation, validation and refinement. The generation component also has a planning stage and the refinement component is designed to be iterative and used together with feedback from the validation component. Distinct from existing methods, the proposed approach focuses on reproducing the issues and retrieves relevant tests to be used in validation. The proposed approach aims to balance efficacy, stability and cost. Evaluations using SWE-bench demonstrate this balance.

**Strengths:**
- Reviewers acknowledge that the proposed method outperforms SOTA and reduces cost. This is in general accepted to be an important milestone and will be of interest to the ICML community.
- Reviewers note that the evaluations use the standard benchmark for this domain, namely SWE-Bench. Having said that, there are mixed signals by reviewers around the design of experiments in terms of supporting the claims of the paper, with three out of four reviewers supporting the experiment design and commenting on the rigor of the evaluations.
- The paper is generally well written, organized and easy to read.

**Weaknesses:**
- There are outstanding concerns around the broad significance of the paper compared to the literature. Reviewers commented on the incremental nature of contributions compared to existing on Agentless and OpenHands and noted that novelty is limited.

**Recommendation:**
- Three out of four reviewers recommended acceptance of the paper with scores of 3,3 and 4. The remaining reviewer recommended a weak rejection (with a score of 2).
- I believe that the paper’s strengths on strong performance on SWE-bench through a combination of human-based and agent based planning approaches and iterative refinement with an emphasis on validation outweigh the weaknesses on limited novelty. I also believe this paper will be of interest to the ICML community as it is a timely contribution advancing agentic approaches for software engineering applications. Therefore, I recommend acceptance of this paper.